**Estimating CO₂ Emissions for 108,000 European Cities**
**Authors:** Daniel Moran[1,*], Peter-Paul Pichler[2], Heran Zheng[1], Helene Muri[1], Jan Klenner[1],   Diogo
Kramel[1] Johannes Többen[2], Helga Weisz[2], Thomas Wiedmann[4], Annemie Wykmans[5], Anders Hammer
Strømman[1], Kevin R. Gurney[6]
Affiliations:
1. Programme for Industrial Ecology, Department of Energy and process Technology, Norwegian University of Science and
Technology, Trondheim, Norway
2. Potsdam Institute for Climate Change Research (PIK), Potsdam, Germany
4. Sustainability Assessment Program, School of Civil and Environmental Engineering, UNSW Sydney, Australia
5. Faculty for Architecture and Design, Norwegian University of Science and Technology, Trondheim, Norway
6. School of Informatics, Computing, and Cyber Systems, Northern Arizona University, Flagstaff, AZ, USA
* Corresponding author: daniel.moran@ntnu.no
## Abstract
City-level CO₂ emissions inventories are foundational for supporting the EU's decarbonization goals.
Inventories are essential for priority setting and for estimating impacts from the decarbonization
transition. Here we present a new CO₂ emissions inventory for all 116,572 municipal and local
government units in Europe, containing 108,000 cities at the smallest scale used. The inventory
spatially disaggregates the national reported emissions, using 9 spatialization methods to distribute
the 167 line items detailed in the National Inventory Reports (NIRs) using the UNFCCC Common
Reporting Framework (CRF). The novel contribution of this model is that results are provided per
administrative jurisdiction at multiple administrative levels, following the region boundaries defined
OpenStreetMap, using a new spatialization approach. All data from this study is available at Zenodo
https://doi.org/10.5281/zenodo.5482480 and via an interactive map at https://openghgmap.net.

## 1. Background
While climate goals are set at the national and international level it is often local governments and
citizens who are most intimately involved in the accomplishment of these goals, and who must adapt
to the implied changes. The European Commission has been clear that cities will play a central role in
reaching European climate goals. As with nation-states, a greenhouse gas (GHG) inventory is the first
step to preparing a local climate action plan (CAP). Cities often use one of the various protocols
available or develop their own methodology to create an emissions inventory. And for good reason -
an inventory informs all levels of municipal decision making, from long-term planning strategies to
infrastructure investments and day-to-day management of building permits. Nevertheless, many local
governments in Europe still do not have a good estimate of their own GHG emissions. Establishing an
emissions inventory is laborious and can be costly for jurisdictions that do not have in-house expertise.
Hence, as the spotlight turns to cities to effect and manage a successful transition to carbon neutrality,
many see the preparation and maintenance of a local emissions inventory as a considerable challenge.
Cities can develop their own inventories using a protocol such Global Protocol for Community-Scale
Greenhouse Gas Emissions Inventories (Fong et al., 2016) a joint initiative of WRI, the C40, Global
Covenant of Mayors, and ICLEI (Kona et al., 2021). An inventory informs all levels of municipal decision
making, from long-term planning strategies to infrastructure investments and day-to-day
management of building permits.
A number of GHG monitoring, reporting, and verification (MRV) solutions have been put forward.
These include sensor networks (both ground and space-based), and a range of accounting and model-
based approaches. No one of these approaches is ideal: they differ in terms of accuracy, precision,
cost, and scalability. In response it has therefore been suggested that MRV efforts should aim at
triangulating true $CO_2$ emissions using a mix of empirical, modeling, and statistical methods (Lauvaux
et al., 2020; Mallia et al., 2020). The model presented here should be seen as one estimate, to be
combined with other estimation approaches and local knowledge, to triangulate towards an
actionable emissions inventory.
One approach for cities to monitor emissions is by using atmospheric measurement of GHG
concentrations and "inverting" that for an emission quantity. These efforts require atmospheric
transport models to translate the atmospheric mixing ratios into surface fluxes of GHGs (Davis et al.,
2017; Ghosh et al., 2021). Concentration measurements can include dense, low-cost sensors (Kim et
al., 2018), high-precision tower-mounted instruments (Turnbull et al., 2019; Whetstone, 2018),
aircraft and satellite-based measurements (Nasa, 2021; Jaxa, 2021; Wu et al., 2020), the EU's CoCO2
and ICOS Cities projects, NASA's OSSE project (Ott et al., 2017)  and/or combinations of all of the
above. By combining these approaches with high-resolution emission data products built using
bottom-up approaches, attribution to emitting source by sector or fuel is possible and has shown good
convergence (Basu et al., 2020; Lauvaux et al., 2020; Mueller et al., 2021).
Many estimates of emissions using techniques independent of atmospheric monitoring have also been
accomplished. These inventory approaches are often described as being either "top-down" or
"bottom-up" (though in fact models may use a combination of these approaches). Top-down models
begin from national statistics, such as national energy use or fuel import statistics, while bottom-up
approaches estimate emissions at the point of combustion or emission release based on deterministic
information (e.g. fuel combustion characteristics, leak rates) and then aggregate these to an implied
national total. The top-down approach uses spatial proxies such as gridded population, nighttime
lights, GDP estimates, and other available spatial proxy variables to allocate national total emissions
across grid cells in each country. Bottom-up techniques often use a mixture of data such as direct flux
monitoring (e.g. powerplant stack monitors), local fuel or utility data, and traffic monitoring.
Several global and country-scale spatially explicit GHG inventories have been developed based on
either bottom-up or top-down approaches. The JRC EDGAR v6.0 (Crippa et al., 2020), ODIAC (Oda and
Maksyutov, 2011; Oda et al., 2018) are well-established examples of global emission data products
but others have been developed (Andres et al., 1996; Andres et al., 2016; Asefi-Najafabady et al., 2014;
Nassar et al., 2013; Rayner et al., 2010; Wang et al., 2013), including some at the national/regional
scale (Bun et al., 2019; Zheng et al., 2021; Jones et al., 2020; Kurokawa et al., 2013; Meng et al., 2014).
A number of these models use nighttime lights data as one input signal (or gridded population datasets
which in turn may be based on nighttime lights), though at least one study has found this is only
moderately predictive (Gaughan et al., 2019).
Spatially explicit bottom up GHG inventories have been accomplished at the regional, national and
urban scale. For example, the US 1 km2/hourly VULCAN $CO_2$ emissions data product (Gurney et al.,
2020a; Gurney et al., 2009; Gurney et al., 2020b) and the Northeast US 1km2 ACES (Gately and Hutyra,
2018) data product. Similarly, work in Poland has achieved similar success (Bun et al., 2010; Bun et al.,
2019). Building/street scale bottom-up efforts have also been accomplished with the HESTIA Project
which has estimated hourly urban $CO_2$ data products in the four US cities (Gurney et al., 2019; Gurney
et al., 2012; Patarasuk et al., 2016; Roest et al., 2020).
Finally, urban emissions have been estimated at the whole-city scale using both top-down and
bottom-up techniques as individual city studies or as collections of urban areas (Ramaswami and
Chavez, 2013; Chen et al., 2019; Harris et al., 2020; Jones et al., 2020; Meng et al., 2014; Shan et al.,
2018; Shan et al., 2017; Zheng et al., 2021; Long et al., 2021) as well as results focused on city results
in England (Baiocchi et al., 2015), China (Liu et al., 2020; Wang et al., 2017), and Europe (Baur et al.,
2015). Many of these studies extend analysis to include Scope 3 or consumption emissions.
Here we provide a new pan-European model estimating emissions at the municipality level (Moran,
2021). This is intended to be useful for cities which have not conducted their own inventory. The
inventory disaggregates the totals from the official national $CO_2$ inventory, summarizing the 167 line
items of the UNFCCC's Common Reporting Framework  (hereafter, CRF) (Ipcc, 2006) into nine
emissions categories. The model identifies up to five levels of administrative hierarchy across 34
European nations including the UK.
This paper proceeds by first situating this contribution with respect to similar work. We then present
the methodology and results, including a pixel and city-level comparison with EDGAR and ODAIC and
a first validation against 43 existing urban emissions inventories assembled by individual cities. We
conclude with a discussion in which we reflect on use cases and next steps.
The JRC EDGAR database, ODIAC, and GCP-GridFED databases are obvious points of comparison to the
model we present in this study. Section 3 presents a conceptual and numerical comparison of these
datasets. The main innovations presented by this model over EDGAR and ODIAC are (a) results are
provided for administrative jurisdictions rather than on a raster grid and (b) the use of OpenStreetMap
is novel. Additionally, our model is targeted to be useful to citizens and local governments, at city level,
by identifying the sources of their city's $CO_2$ emissions. This influences some of our modeling
approaches, such as emissions attribution from ships and planes to ports and airports rather than
along their physical voyage tracks. But it is the provision of ready-to-use results at the city, county,
and state level across Europe which we believe is the core contribution of this database.
The method described here is intended for creating an inventory of direct emissions. It is worthwhile
to recall the distinction between scope 1, 2, and scope 3 emissions inventories as defined in the WRI's
Greenhouse Gas Protocol nomenclature (WRI et al., 2014). An inventory of direct emissions is called a
scope 1 inventory, a territorial emissions account, or a production-based emissions account (PBA). A
scope 2 inventory will be largely identical to a scope 1 inventory but reallocate the emissions from
electricity production to the location where that electricity is directly used. A scope 3 inventory, also
called a footprint or a consumption-based account (CBA), will further expand the scope and attribute
to consumers all emissions associated with imported goods and services produced domestically or
abroad, and emissions associated with waste exported outside the jurisdictional bounds. For urban
areas with little production and much consumption, scope 3 emissions can be substantial: studies
estimate that for many urban cores their scope 1 emissions are 30-50% of their total scope 3 footprint.
Scope 3 inventories are estimated using trade and supply chain databases and rely on robust (i.e. well-
modeled or empirically validated) scope 1 inventories as a starting point.  There is an active community
working to prepare Scope 3 assessments at the city level (Chen et al., 2019b, a; Guan et al., 2020;
Heinonen et al., 2020; Minx et al., 2013; Moran et al., 2018; Pichler et al., 2017; Ramaswami et al.,
2021; Wiedmann et al., 2021; Zheng et al., 2021b).

## 2.  Methods

The approach presented here spatializes the national emissions inventory using activity data from
Open Street Map (OSM), the EU's Emissions Trading System registry of point source emitters, and
traffic data for airports. This method sums to a national total equal to the national inventory,
generates results as both a gridded dataset and per administrative unit and preserves detail on the
sources of emissions. The intention is to best locate emissions to where they physically or legally occur.
As the spatial resolution of the inventory increases an interesting consideration emerges, namely that
there is some discretion in where emissions should be spatially located. The emissions for a passenger
ferry for example could be spatially located over water where they physically occur, at the office of
the ferry company which is legally responsible, at an industrial harbor where the boat takes on fuel,
or at the passenger terminals where it traffics. At larger grid cell sizes these four locations are more
likely to share the same grid cell, but with highly resolved models this becomes a modeling choice.
Our choices on such decisions are documented in the relevant section of methods which describes
each emissions category, but as a general principle we opt to locate emissions where it makes most
sense for communication and outreach by those using the results, where policy tools are easiest to
apply, or where they physically occur, in that order.
Scope of coverage: The model is currently built for the year 2018. This is the most recent year for
which official national inventories were available from EUROSTAT when the model was assembled.
The list of countries covered is provided in the Results section of this paper. The UK is included in the
model. Regarding the impact of the UK's exit from the EU, we anticipate this will not substantially
reduce the ability to use this model for the UK, since the UK has established its own UK ETS and, we
presume, will continue to publish an emissions inventory in CRF (the Common Reporting Format)
format. This study focuses only on $CO_2$ emissions; other greenhouse gasses are not included. In each
relevant section of the Methods a discussion is included about how the model could be extended to
handle other GHGs. One rationale for this choice is that the second largest GHG, $CH_4$, is heavily driven
by agricultural activities and rogue emissions and these are some of the hardest to accurately
spatialize. Furthermore, the intention in this study is to focus on fossil fuel use and not short-cycle
carbon such as emissions related to land use and agriculture. Therefore, the model does not include
emissions from land use, land-use change, and forestry (LULUCF). The choice to exclude these from
the model was based on considerations including (a) estimates of total LULUCF emissions are often
poorly constrained, (b) they are difficult to spatialize accurately, (c) local government policy have
fewer immediate policy options for managing these emissions, (d) national climate targets often
exclude LULUCF emissions, (e) there are diverse approaches to accounting for LULUCF and carbon
sinks, leading to significant variability (Grassi et al., 2018; Petrescu et al., 2020).
The model assembly procedure can be summarized as follows. Further detail and discussion on each
aspect is provided in the following subsections. First, emissions which can be attributed to point
source facilities reporting under the ETS are separated from the national inventory. ETS-registered
emissions are geolocated at the street address registered for that permitholder. In the cases where
the location of emissions differs from the registered address (e.g. offshore oil activities, or some
company activities) this approach can still be rationalized since (a) physically locating all facilities which
are not at their mailing address will be difficult, and (b) legally, the control of the emissions is likely at
the registered address, so there is sense in calling attention to emissions which are controlled from
there. Emissions from vehicles are apportioned equally to fuel stations as located in OSM. The model
amortizes total national vehicle fuel use evenly across all fuel stations, though this will not correctly
capture subtleties such as fleet and trucking-only fuel depots, nor differentiate between small (1-2
pump) stations and large filling stations with multiple pumps. Emissions which are associated with
buildings (heating and cooling, construction, and light commercial activity), plus the residual industrial
emissions which cannot be attributed to ETS sources, are apportioned equally onto all buildings
registered in OSM. (OSM does allow buildings to be tagged with extended attributes such as floor size,
stories, and use, but in our investigations <1% of buildings use these attributes so for now we have
not attempted to utilize those fields.) Emissions from marine bunker fuels are apportioned equally to
harbors as located in OSM (note that diesel fuel for small vessels will be treated as vehicle fuel).
Emissions from aviation bunker fuel are spatialized onto airports proportional to the volume of
passenger traffic handled at each airport, as reported by Eurostat. Fugitive emissions and emissions
from petroleum byproducts are spatialized equally across national refineries and associated oil
storage facilities. $CO_2$ emissions from farming and forestry are apportioned to farmed areas as located
in OSM (these are based on the EU CORINE land use map). Emissions from trains are mapped to
passenger train stations.
Figure 1 displays the total emissions covered in the model, excluding of LULUCF and carbon sinks,
grouped according to the methods used to spatialize those emissions, and color coded according to
the approximate level of difficulty, or degree of uncertainty, of that spatialization, with greyer colors
representing more easily spatialized emissions and brighter colors indicating emissions categories
which, in the authors' experience, are more difficult to confidently spatialize.

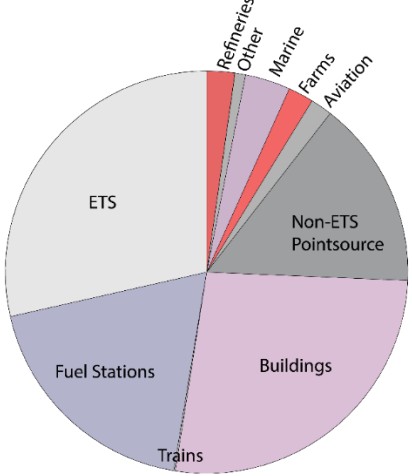

*Figure 1: Composition of emissions across the 34 European countries covered. ETS shows the volume of emissions associated*
*with ETS-registered point source emitters; fuel stations show emissions from vehicles; the 'buildings' category comprises*
*emissions from building heating, cooling, construction, and light commercial activity. Non-ETS point source emissions is a*
*residual category representing the difference between industrial emissions as reported in the national inventory and the sum*
*of emissions reported by facilities participating in the ETS. Nearly half (42%) of these occurs in Turkey, which as of publication*
*does not participate in the ETS, but this discrepancy is also observed in large emitters like Germany, France, the UK, and*
*Poland. These residual emissions are spatialized using OSM records instead of ETS addresses.*

## a. Mapping point source emissions regulated by the EU Emissions Trading System

The EU's Emission Trading System (ETS) requires large point-source emitters to report emissions and
report an address for every permitholder. A geolocation API was used to translate these addresses
into latitude-longitude coordinates. While for many facilities the address where the emissions are
legally controlled is the same as the facility's physical address, or in a nearby town, in some cases the
two locations can differ more substantially (emissions from Norwegian offshore activities are largely
legally controlled in the city of Stavanger, for example).  The emissions associated with ETS permitted
facilities are then subtracted from the CRF inventory thus leaving fewer total emissions remaining to
be spatialized. The allocation of CRF emissions to ETS facilities is done as follows. For a number of CRF
sectors (for example, "Fuel combustion in manufacture of iron and steel" (1.A.2.A)), some or all of the
sector's emissions are attributable to ETS facilities. We constructed a priority-ranked concordance
table to determine which CRF emissions are already covered by ETS-registered permits. Normally the
ETS-reported emissions for a given activity are less than or equal to the CRF-reported emissions for
that category and there is only a small residual between the CRF-reported value and sum across
pertinent ETS permits, however in some cases this residual is substantial.
The mapping between ETS categories and CRF categories is not always one-to-one. For example, the
ETS uses the code "24: Production of pig iron or steel". These facilities may correspond to the CRF
activities, Fuel combustion in manufacture of iron and steel (1.A.2.A), Iron and steel production (2.C.1),
or Ferroalloys construction (2.C.2). In our ranked concordance matrix approach, a rank of 1 is given to
the first CRF activity, a rank of 2 is given to the second CRF activity, and a rank of 3 is given to the third
CRF activity. The emissions from those ETS facilities from code 24 are first attributed to the rank 1 CRF
activity until it is sated, then excess ETS emissions are assumed to come from the rank 2 activity until
that volume is sated, the same for rank 3, and so on. Using the above example that could mean that
all emissions under the first two CRF categories would be fully attributed to ETS iron and steel facilities,
and a portion of the emissions under rank 3, Ferroalloys construction (2.C.2), which cannot be
attributed to ETS facilities, would remain to be spatialized.
In some cases it is unclear what the ranking of CRF activities should be. For example after allocating
ETS emissions from "production of lime, or calcination of dolomite/magnesite" (ETS category 30) first
to  lime production (2.A.2) and secondarily to glass production (2.A.3), should excess ETS facility
emissions from code 30 best be attributed to Cement production (2.A.1), Fuel combustion in
manufacture of non-metallic mineral products (1.A.2.F), or Fuel combustion in other manufacturing
industries and construction (1.A.2.G)? In this case the last three sectors are sated in smallest-to-largest
order until no ETS emissions remain to be allocated. The rationale for the ascending sort order is that
larger CRF categories will be easier to spatialize using other methods. In the earlier example of
aluminum production, any surplus reported in ETS which exceeds the CRF reported aluminum
production emissions is then assigned to the rank 2 CRF category of "Fuel combustion in other
manufacturing industries and construction", decreasing the amount of emissions in that CRF category
which remain to be spatialized. We also note that not all facilities use the expected ETS activity code.
For example we have observed some fertilizer plants reporting emissions under ETS activity code 42
"Other Bulk Chemicals" instead of activity 41, "Ammonia production". Such misattributions can
introduce distortions in the model results. To characterize the impact of these distortions the
allocation of ETS emissions through the ranked priority allocation system into CRF would need to be
followed manually in detail.
After linking ETS-reported emissions to the national inventory, the remaining CRF-reported emissions
are spatialized using the methods described as follows.

## b.  Vehicles

These are emissions from the following five CRF categories
1.A.3.B.i   Fuel combustion in cars
1.A.3.B.ii  Fuel combustion in light duty trucks
1.A.3.B.iii Fuel combustion in heavy duty trucks and buses
1.A.3.B.iv  Fuel combustion in motorcycles
1.A.5.B Mobile fuel combustion sectors n.e.c.
These emissions are specialized according to the location of vehicle fueling stations as documented in
OpenStreetMap. We make the assumption that the number of vehicle fuel stations in an area is
proportional to the volume of traffic served. This is a simplifying assumption and it is clearly
communicated in the model presentation. In future development of the model, localizing vehicle
emissions will be a top priority (for comparison, we note the Carbon Monitor project's use of TomTom
live vehicle location data to spatialize traffic.(Liu et al., 2020)). This approach assumes that every fuel
station supplies a similar level of vehicle traffic. It could be the case that some stations are small single
pump gas stations while others are large facilities, for example such as located along a major highway
rest stop. To address this one future solution could be introduce better road traffic estimates. While
traffic load estimates are available for some roads, these estimates tend to be for only a few dozen
specific highways. Fu and colleagues (Fu et al., 2017) proposed a method using neural networks to
estimate vehicle flow on every road using OSM data and gridded population models. (Osses et al.,
2021) recently prepared a high-resolution map of emissions from vehicles in Chile. Better modeling
vehicle traffic, not only fuel station availability, would make the model more accurate in spatially
estimating vehicle fuel emissions. Another potential solution would be to identify data on fuel station
volume, e.g. sales estimates or number of pumps installed, but this may be challenging in practice. A
second assumption is that every station serves a homogeneous mix of vehicles. It may be the case that
some stations serve a specific fleet, for example a city bus fleet, and better identifying the mix of
vehicles served by each fuel station would allow the above five emissions categories to be more
precisely spatialized. Insofar as electric car adoption drives some fuel stations to close the model will
reflect lower vehicular emissions in areas with more electric vehicles. An interesting note is that in
some urban centers light truck traffic is suspected to be a larger emission source than passenger
vehicles. Better distinguishing types of traffic and vehicles would be useful for helping guide
decarbonization plans that are most appropriate for various areas.

### c.  Trains

Trains are a relatively minor source. Emissions for Fuel combustion in railways (1.A.3.C) were
spatialized using passenger train stations as reported in OSM. Every train station was allocated an
equal share of the total emissions. A limitation of this approach is that it may be that not all train
traffic is equally fuel-intensive: some individual trains or sections of the rail network could be fully
electrified and other areas not. Another limitation is that the method allocates total train emissions
(both passenger and cargo) equally across passenger stations, yet passenger stations are not all
equally used, and cargo train activity would be more appropriately localized at freight yards. Reporting
train emissions at passenger terminals does service a communicative value as it reminds viewers that
train traffic is not entirely emissions-free.

### d.  Buildings

In the following categories, only a portion of the emissions can be spatialized to ETS locations, but
there remain emissions which must be spatialized onto buildings:

1.A.2.G Fuel combustion in other manufacturing industries and construction
1.A.4.A Fuel combustion in commercial and institutional sector
1.A.4.B Fuel combustion by households
1.D.3 Biomass - CO2 emissions (memo item)
2.D.3 Other non-energy product use

The largest shares of these remaining emissions are driven by building heating and cooling and fuel
combustion by light industry and construction.
Correctly spatializing these emissions associated with buildings is a substantial challenge. OSM is
sometimes known as Open Buildings Map since the database actually contains more buildings than
streets. The OSM dataset reports an extensive number of buildings, but little data is available to
characterize each building. OSM does not record all buildings. In many areas, including small towns,
only a street address is marked but there is no point or polygon data indicating what is built at that
address. While it might be possible to obtain maps of all buildings from national cadaster agencies,
part of our intention in the model is to develop methods which are replicable across other countries
and not rely on single-country datasets. Of the buildings recorded in OSM, only a small percentage (1-
5%, depending on country) contain any information characterizing the building such as number of
floors, main usage activity, building material type, or building age. Some recent offerings which
provide building footprints (e.g. products from Maxar or Predicio Building Footprint Data, free
offerings from Bing / Microsoft, and academic initiatives such as coordinated through spacenet.ai)
could be used to identify at least building footprint size, and potentially height or construction
material.
The approach used in the model is to apportion all of the emissions associated with buildings equally
among all buildings and registered street addresses in each country. It is important to recall that for
buildings heated by electricity, $CO_2$ emissions associated with electricity production will be located at
ETS-registered power plants. As noted above, there is a paucity of information available by which we
could further characterize building size or use.

### e. Aviation

Total emissions associated with kerosene used for aviation fuel (the sum of the CRF categories "Fuel
combustion in domestic aviation (1.A.3.A)" and "International aviation (1.D.1.A)") reported by EU
member states and calculated compliant with IPCC 2006 guidelines (Maurice et al., 2006). These
emissions are attributed to airports proportionally to total passenger kilometers (pkm). Fuel use from
military aviation is excluded.
Total pkm are derived from the combination of EUROSTAT statistics of route traffic and passenger
traffic per airport. This procedure is preferred over an attribution based solely on total passenger or
flight numbers, since we here implicitly incorporate information on both the flight length and aircraft
size. These parameters are two major drivers for fuel consumption and emissions (Yanto and Liem,
2018).


### f. Farming Activity

The CRF uses the following three categories for farming-associated activities:
1.A.4.C Fuel combustion in agriculture, forestry and fishing
3.G Liming
3.H Urea application
The largest of these, category 1.A.4.C, is challenging to spatialize for two reasons: First, the inclusion
of fishing activity means emissions in this category overlap with emissions in marine traffic. To handle
this, emissions from fishing would have to be estimated, removed from this amount, and spatialized
separately. Even then, the remaining emissions from fuel combustion in agriculture and forestry would
still be difficult to spatialize. Second, we have not been able to identify a suitable dataset to use to
divide and appropriately spatialize forestry as distinct from farming.
Our approach is to map these collected emissions onto locations of farmland as identified by the EU's
CORINE land-use dataset, which is already incorporated into OSM. The above emissions were evenly
allocated to the centroid points of all polygons tagged as farmland from CORINE. This approach will
not correctly spatialize emissions associated with forestry. Also, this approach allocates the emissions
evenly across every polygon tagged as farmland, regardless of the size of each patch. A future
improvement could be to weight this allocation by patch size and thus assume every hectare of
farmland is equally emissions-intensive to manage, or to introduce activity-level data for agriculture,
such as integrating maps of dairy cattle operations (Neumann et al., 2009) or similar.
As discussed in the introduction, and in section 10 below on short-cycle carbon, currently the model
intentionally excludes emissions from land use, land use change, and biotic processes such as cattle
digestion and manure handling.
The following categories in the CRF report also relate to farming:
3 Agriculture
3.1 Livestock
3.A Enteric fermentation
3.B Manure management
3.C Rice cultivation
3.D Managed agricultural soils
3.E Prescribed burning of savannas
3.F Field burning of agricultural residues

## g. Marine

Emissions from the maritime sector are part of international bunker fuel emissions together with international aviation. In both cases, emissions are calculated as part of the national GHG inventories but not included in national totals.

Emissions in this sector are comprised of the following CRF emissions categories:

  1.A.3.D 4 Fuel combustion in domestic navigation
  1.D.1.B 4 International navigation

This covers tank-to-wake emissions that stem from fuel combustion. Total fuel consumption is calculated by a top-down assessment based on annual sales of bunker fuel in each country, comprising marine gas oil (MGO) and heavy fuel oils and distillates (HFO), and geospatially distributed across the 888 ports.

Port-allocation of bunkered fuels is based on the total transport work for berth-to-berth ship voyages, as obtained from IHS Markit, totaling 773 000 port calls. Ship voyages are combined with their ship's respective average fuel consumption as reported by shipowners to the European Union's emissions monitoring scheme (the EU MRV, Monitoring, Reporting and Verification), given as kilograms of fuel per nautical mile. This covers all vessels operating in EU ports above 5000 GT, totalling approximately 11 000 vessels. The distance covered with each voyage is calculated by applying the Dijkstra's algorithm (Dijkstra, 1959) to find the shortest path between two ports, followed by a curve smoothing process by the Ramer–Douglas–Peucker algorithm (Douglas and Peucker, 1973; Ramer, 1972). The average fuel consumption and distance sailed is used to estimate total bunker demand at the port level, by weighing the national reported bunker sales. This approach is expected to be gradually replaced by the bottom-up emission inventory provided by the MariTEAM model (Kramel et al., 2021).


This assessment does not include leisure crafts, considered negligible in comparison to cargo vessels,
neither does include warships, naval auxiliaries, fish-catching or fish-processing ships that are exempt
of reporting their activity to MRV.


**h. Other**
There are some emissions which are difficult to spatialize. These are:

1.C Transport and storage of CO2 (memo item)
2.A.4 Other process uses of carbonates
2.D.1 Lubricant use
2.D.2 Paraffin wax use

In the model these emissions are included in and spatialized using the same strategy as emission from
buildings as described above.

**i. Refineries**
The following CRF emissions categories are associated with oil refineries and fossil fuel infrastructure:

1.B 2 Fuels - fugitive emissions
1.B.1 Solid fuels - fugitive emissions
1.B.2 Oil, natural gas and other energy production - fugitive emissions
2.B.8 Petrochemical and carbon black production

Carbon black, item 2.B.8, used to produce black ink, is a byproduct from fracking at refineries. Fugitive
emissions (1.B.2) are by their nature difficult to spatialize (Plant et al., 2019). A number of studies in
California have tried to characterize fugitive emissions from the ageing oil wells and modern fracking
equipment in the region (Hsu et al., 2010; Rafiq et al., 2020; Townsend-Small et al., 2012; Wennberg
et al., 2012). In our model all fugitive emissions are attributed evenly across refineries and associated
storage tanks as located in OSM. The fugitive emissions are apportioned equally among the buildings
tagged [industrial=refinery] or [industrial=oil] in OSM. This approach has the disadvantage of not
correctly spatializing fugitive emissions at the various wellheads, pumping and storage locations
where such emissions physically occur, but has the advantage of attributing fugitive emissions to
refineries so that policy planning can recognize that fossil fuel creates emissions both when it is
combusted but also during its production. This approach follows the guiding philosophy of locating
emissions where they best connect to the relevant policy discussion.

**j. Land Use, Forestry, Stock Change, and Waste (Short-cycle carbon)**
Our model is focused on reporting $CO_2$ emissions from fossil fuel combustion and industrial processes.
We explicitly set aside so-called "short cycle carbon", that is, carbon which is already in the biosphere
stock. We limit the model to focus on emissions of carbon taken from the fossil stock.
Carbon put into sinks (under CRF table 5 - Waste), either natural (terrestrial, aquatic, or marine) or
manmade (e.g. timber construction or paper or biomass put into landfill) sinks is not spatialized or
included in the results. Negative emissions from carbon capture and storage facilities are presently
excluded from the model.
$CO_2$ emissions from CRF category 4, encompassing land use, land use change, and forestry, are also
not included. Our intention is to spatialize fossil fuel combustion associated with agriculture and

forestry but not emissions associate with landscape-scale soil and biotic processes. We reason that such landscape-scale emissions are both large, and very challenging to address using locally available policy tools. Including them in a city-oriented plan, particularly in rural municipalities, could lead to a situation where the results are heavily dominated by an emissions category with few viable solutions.

In future iterations of the model it may be preferable to allow users to easily include or exclude the emissions in the model results. Currently our model does not include direct $CH_4$ emissions from cattle digestion and manure fermentation. This is a substantial emissions category with some remediation options so it may be useful to include this in a future iteration of the model.

Another detail in this category is sewage treatment and landfills. These act as both sources and sinks of carbon. It is unclear whether net emissions from sewage plants and landfills are included inside the CRF category "Long-term storage of carbon in waste disposal sites" (5.F.1) or included in another category. As category 5.F.1 is not included in the model, if net emissions from sewage are included in this category those emissions will not be included in the model. Quantifying emissions associated with sewage treatment and local landfills would be an improvement to the model.

## 3. Benchmarking

We do not intend here to provide an exhaustive survey of available spatial emissions models. Here we only compare the OpenGHGMap model with some widely used global-level models. A full comparison of spatial emissions models, including several strong single-country models, would be a valuable contribution to the field, but is not within the scope of the present paper. For one such comparison we refer to (Hutchins et al., 2017).

Table 1 provides an overview and comparison of OpenGHGMap with ODIAC (Oda and Maksyutov, 2011), JRC's EDGAR (Crippa et al., 2020; Crippa et al., 2019), and the Global Carbon Project's GCP-GridFED (Jones et al., 2020) spatial emissions models.

| | Resolution | Itemization | Temporal | Results by jurisdiction | Scope | Method synopsis |
|---|---|---|---|---|---|---|
| ODIAC | 1km | Total emissions | Monthly | Country | Global | Spatialize national emissions using nighttime lights and power plant locations |
| EDGAR v6.0 | 0.1° (11km at the equator) | 31 IPCC CRF categories | Up to hourly | Country | Global | Collected activity-level data sources (e.g steel industry, FAO for farming activity, ship and flight tracks) |
| GCP-GridFED | 0.1° (11km at the equator) | Total emissions, per 5 fossil fuels | Monthly | Country | Global | National totals from GCP, spatialized using EDGAR |
| OpenGHGMap (our model) | Point-source, 1km grid, or per municipality | 9 categories | Annual | Country, State, County, Municipality, facility | Europe | Spatialize national emissions using activity data from OpenStreetMap |

*Table 1: Comparative overview of several spatial emissions datasets.*

Comparison to EDGAR, and GCP-GridFED which uses EDGAR's spatialization layer: At the time of writing, the report with the methodology used for the EDGAR v6.0 has not been published. Based on the data sources mentioned at the EDGAR website it appears that activity-level data has been obtained for various industrial activities (e.g. farming, fertilizer production, steel refineries, electricity

generation), and plane and ship emissions are mapped to voyage tracks, but it is not published how emissions from buildings, light commercial activity, and vehicles are spatialized, except the GHS-POP gridded population dataset is mentioned. Since OpenGHGMap uses ETS facility-level data to map industrial emissions (an advantage afforded by its Europe-only focus) it may be that the two models will come to similar results for mapping industrial emissions since presumably the activity-level datasets for industry used by EDGAR will be largely identical to the facility-level data from ETS. If EDGAR uses population density as a proxy to map vehicle and building emissions, this is a slightly different approach than OpenGHGMap's use of fuel stations and building locations from OSM.

Compared to ODIAC: The original ODIAC was a ground-breaking project and introduced the approach of using power plant locations and nighttime lights as a proxy for emission activities. Since that project, more recent projects have introduced more proxy variables and activity inventories. In our results comparison (below) the ODIAC results still agree, but ODIAC does not present results with sector/activity detail which is important for further insight and to guide action.

In addition to this conceptual comparison of methods we also compare the numerical results. To compare the results of the OpenGHGMap model to ODIAC and EDGAR v6.0 to the OpenGHGMap model was rasterized to a 30″ (arcsecond) raster (approximately 650 $m^2$ cells at 45° latitude) to permit a direct cell-level comparison across emissions models and the GHS-POP gridded population model. The EDGAR dataset version is v6.0, data year 2018, with a native resolution of 0.1° (360″) before re-gridding. For ODIAC the model version is 2020, with data for 2018, with a native resolution of 1$km^2$ cells. The three modeled inventories report slightly different totals for total European emissions. This is due (a) to differences in emissions categories covered, (b) for ODIAC, the monthly allocation, and (c), for EDGAR, the fact that in EDGAR aviation and marine emissions are spatialized over ship and flight traffic routes rather than allocated to grid cells in the country. For this initial cross-model comparison, the three datasets were normalized to include only grid cells covered by all three models and then by normalizing the total emissions across the three models so that we compare solely the spatial allocation. This is a simplified method for cross-model comparison and leaves considerable scope for future work on cross-model comparison. Our main aim here is to document this new model and conduct a preliminary validation, not conduct a robust cross-model comparison.

The cross-model cell-level comparison (Figure 2) shows the degree of convergence between the OpenGHGMap and the EDGAR model. The OpenGHGMap reports more cells with low (<100 t CO2/yr) and very high (>1000t CO2/yr) emissions. The OpenGHGMap model also reports higher cell-level variability than does ODIAC: the ODIAC model reports most cells have emissions in the range of $10^2$-$10^4$, whereas the OpenGHGMap model reports cells with a range of $10^1$-$10^5$ t $CO_2$/yr. This could potentially be an artefact due to aggregation of ODIAC. The ODIAC model is natively provided at 1$km^2$ resolution, corresponding to a cell size of 0.07-0.04″ depending on latitude, and it could be that the aggregation to 30″ cells for the purpose of comparison has masked higher variability within the 30″ grid. Another hypothesis is that this homogeneity is due to ODIAC's use of nighttime lights data, and that while illumination is relatively homogenous across urban and peri-urban areas, the emissions within similarly lit areas can be starkly different. Another noteworthy feature is that OpenGHGMap reports many more areas with low (<100t) emissions compared to both EDGAR and ODIAC. One hypothesis is that this is related to the method of spatializing emissions from vehicle fuels to fuel stations. Since fuel stations often are spaced >650m apart, especially in rural areas, this could result in many pixels in rural areas being assigned zero fuel emissions. As discussed elsewhere, the decision to localize vehicle emissions at fuel stations was a deliberate design choice in this model. Other models may choose to localize these emissions on roads, or pro-rate them across a gridded population map on a per-capita basis.

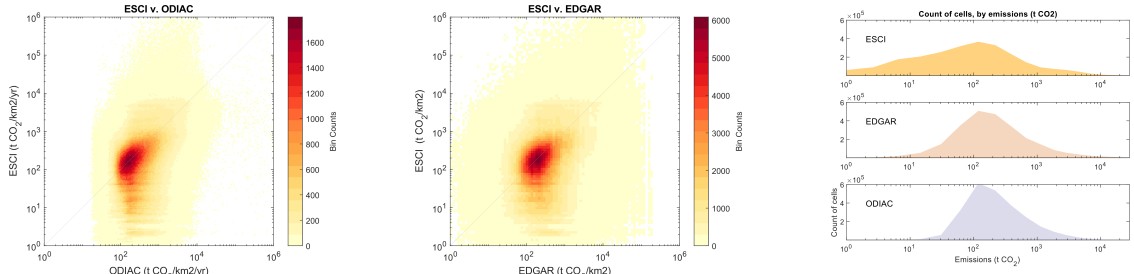

*Figure 2: Emissions per standardized grid cell, cross-model comparisons, and frequency analysis. Compared to the ODIAC*
*dataset (panel a, c), OpenGHGMap reports higher cell-level variability ranging from $10^1$ to $10^5$ t $CO_2$/yr, while ODIAC reports*
*most cells in the range of $10^2$-$10^4$ t $CO_2$/yr. Compared to the EDGAR v6.0 dataset (panel b, c), the OpenGHGMap dataset*
*reports more cells with small (<$10^2$ t $CO_2$) emissions and fewer cells with high (>$10^4$ t $CO_2$) emissions. The OpenGHGMap*
*dataset reveals a higher variability in emissions per cell than do other models.*

Next, we converted the administrative region definitions from OpenGHGMap to a raster map
compatible with the EDGAR v6.0 and ODIAC gridded datasets and we compared the results aggregated
by administrative level across the models, at the city level (i.e. by city) across the models. We
compared results both at the city level, i.e. at the highest level of regional detail per country, and at
the county level, i.e. the administrative level one step above that. These results are presented in Figure
524   3.


(a)                                             (b)

(c)                                             (d)

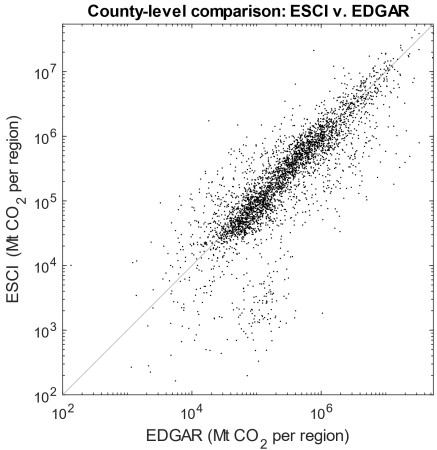


Figure 3: Cross-model comparison of $CO_2$ emissions per city (using the finest level of regional detail) and per county (using the next-finest level of regional detail per country).


Currently no methodology has been developed to quantify uncertainty in the model. In addition to being technically challenging, it is difficult to quantify uncertainty in any single portion of the model, much less the whole. Even if the national inventory or ETS inventory are taken to be 100% reliable, errors and biases introduced during the various steps of spatializing these emissions are difficult to quantify. Developing a strategy for parameterizing reliability of model results would be a valuable next step in the research. Previous studies which have investigated techniques for parameterizing uncertainty in gridded spatial proxy models could be useful (Andres et al., 2016; Bun et al., 2010; Hogue et al., 2016; Hutchins et al., 2017; Woodard et al., 2014).

538

**Validation against city inventories**

The main objective of the OpenGHGMap database is to provide easily accessible estimates for GHG emission inventories at the municipal level to assist local governments in developing more detailed inventories or in developing their own climate action plans (CAP). We compare our OpenGHGMap estimates for external validation with existing municipal GHG inventories compiled from a variety of sources in the 343 Cities dataset (Nangini et al., 2019). These emissions inventories are largely self-reported, of varying quality, and follow different protocols, but still provide the most concrete point of comparison for our Scope 1 emissions estimates at the municipal level. In total, Scope 1 emission values for 44 European cities can be found in the database, which are compared to the OpenGHGMap estimates in Figure 4.

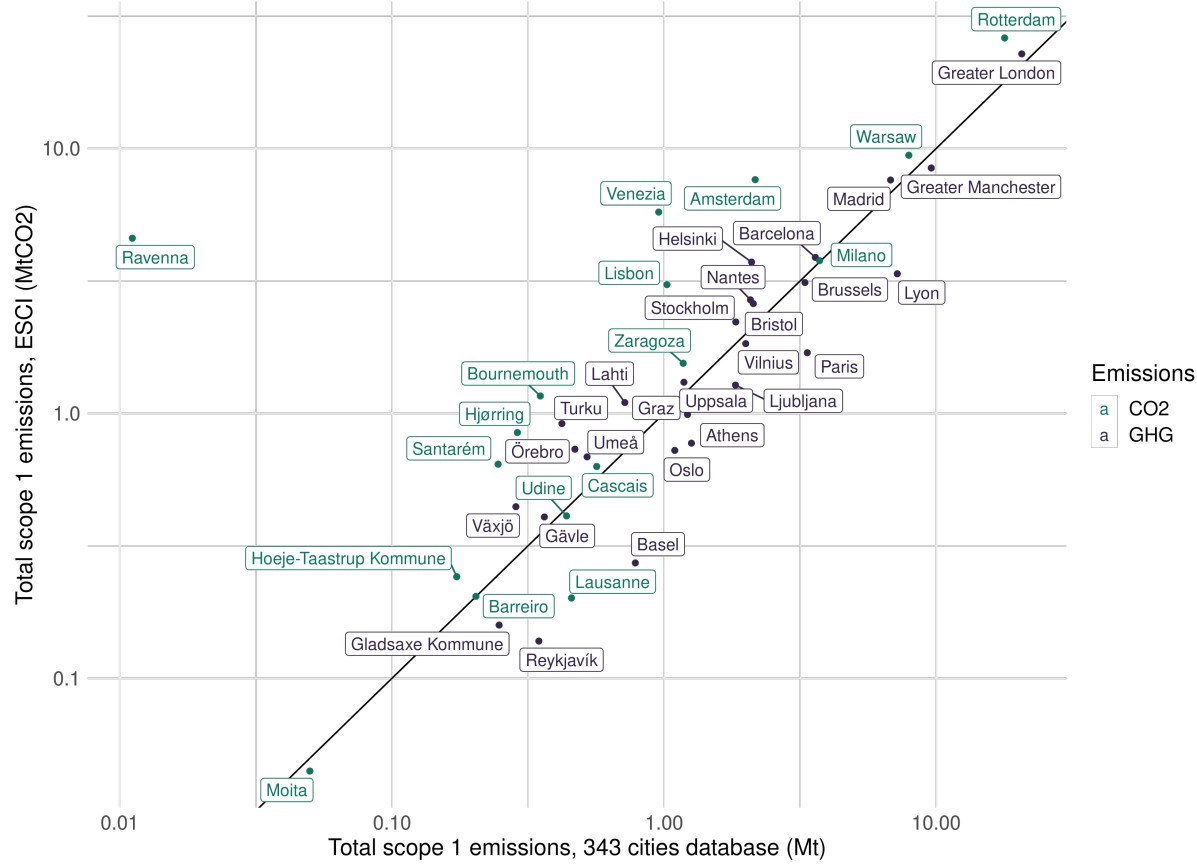

549

*Figure 4: Comparison between OpenGHGMap results and the community level emissions inventories of 44 European cities. Color coding is used to indicate whether the city self-reports CO2 or GHG (CO2eq) emissions. Since OpenGHGMap reports only CO2 emissions this is limited to an indicative comparison, not a precise comparison.*

The figure shows very high agreement (Pearson correlation coefficient 0.937), despite the different methods and timing of the city inventories (emission years between 1994 and 2016 with a median of 2013). Only Ravenna, Italy, differs by several orders of magnitude, but the value in the 343-city database is not realistic (11ktCO2 for a population of 150000 is unrealistically low).

## 4. Main Findings

### a. Results overview

An overview of the results for Europe is shown in Figure 5. The results are presented both in absolute and per capita terms. Some noteworthy features are the high emissions in coastal Netherlands, associated with marine activity, and the high emissions from Gotland island in the Baltic sea, driven by one large cement facility there. Emissions in France are remarkably concentrated into a few, primarily coastal, cities.

One limitation which must be kept in mind when looking at the results at the municipal level is that municipalities vary in size between countries. In continental Europe municipalities are quite small while in the Scandinavian countries the most local administrative units are relatively large and thus aggregate more emissions and are more visually prominent. For some analyses, gridded maps, where the spatial unit of analysis is consistent, are preferable to political maps.

Population per administrative area was estimated by overlaying the administrative boundary on the GHS-POP gridded population map. Gray areas indicate areas where no model results are available. In

some cases (as seen for example in Ukraine and Romania) the administrative regions at that level are
not exhaustive.

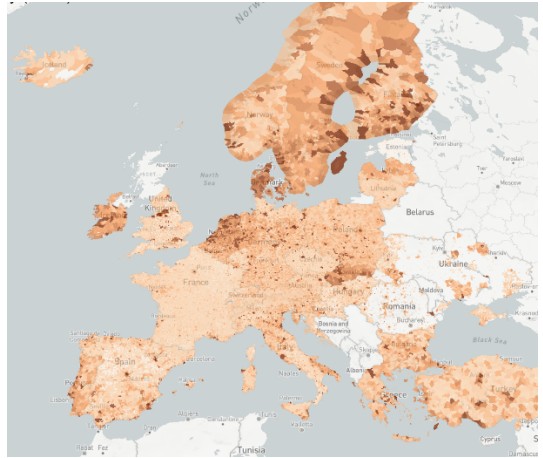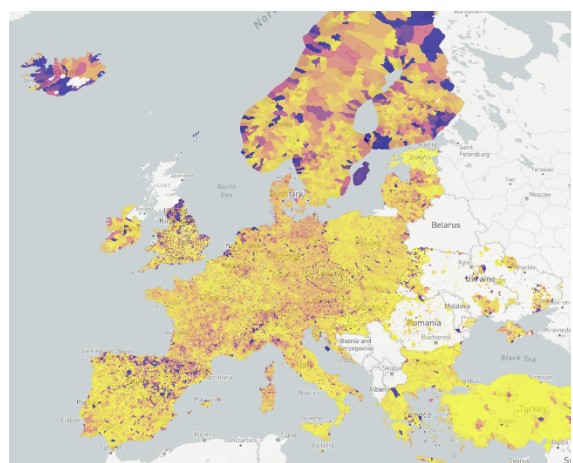


Figure 5: OpenGHGMap.net website screenshots. CO2 emissions per municipality in absolute terms (left panel)
and per capita terms (right panel). Darker colors (browns, purples) indicate higher emissions (absolute values
can be found at the website http://openghgmap.net).

In many countries, emissions are remarkably concentrated in a few regions. As seen in Figure 6, in 21
of the 34 countries assessed, >30% of national emissions arise from ten municipalities. This implies
that focused changes in a few political regions could contribute substantially to achieving national
reduction targets.

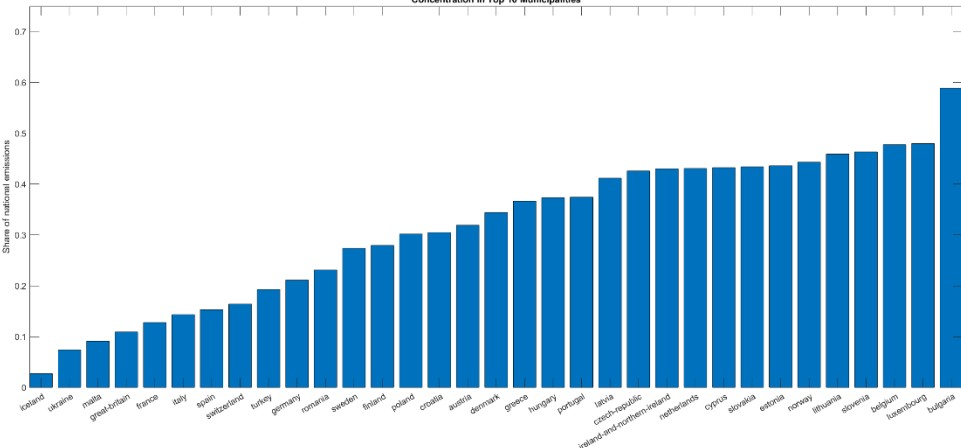


Figure 6: share of national emissions arising from the top 10 emitting municipalities (or smallest finest
administrative distinct) in each country. (Liechtenstein is not shown because the country only has 11
municipalities.)

The important role of high-emitting municipalities is seen at the European level as well. Figure 7
presents a Lorenz curve showing the contribution of municipality to the total European emissions. A
striking degree of concentration is visible, with 10 municipal regions across Europe driving 7.5% of
emissions, 100 driving 20%, and the top 10 cities in each country collectively driving 33.4% of total
European emissions. These highest-emitting regions are not necessarily the most populous, since in
many cases outlying industrial facilities are major drivers of emissions.

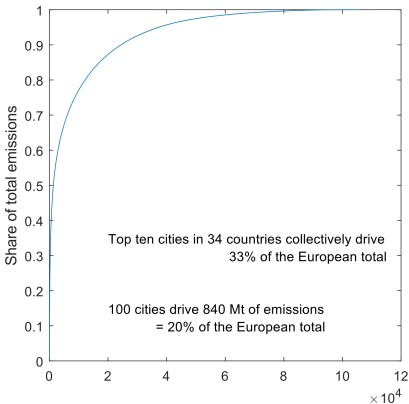


Figure 7: Lorenz curve showing cumulative contribution to total emissions from each municipality.


## b. Case study of Norway


To demonstrate the results provided by the model we investigate Norway as a case study. In Norway
there are just two levels of administrative hierarchy: counties (*fylke*) and municipalities (*kommune*),
corresponding to the NUTS-2 and NUTS-3 levels respectively. This is a relatively simple configuration;
for many European countries the System of administrative hierarchy is complex and deeply historical.
For example in Germany some cities are peers with states and the administrative configuration is
slightly different between states (in some states there is a level 7 administrative subdividision while
in other states there is not); In Switzerland not all cantons use subdivisions; and in some places statistic
agglomerations of areas, such as capital cities with their suburbs, maybe more relevant than the
judicial regions. Our model provides results at all administrative levels in a country as defined in OSM.
There are up to 10 levels available (we do not include level 11, which is for neighborhoods and
parishes) and most countries use between 2 and 5 levels.
*The        results        for        Norway        at        the        NUTS-2        (fylke)        level,        (*

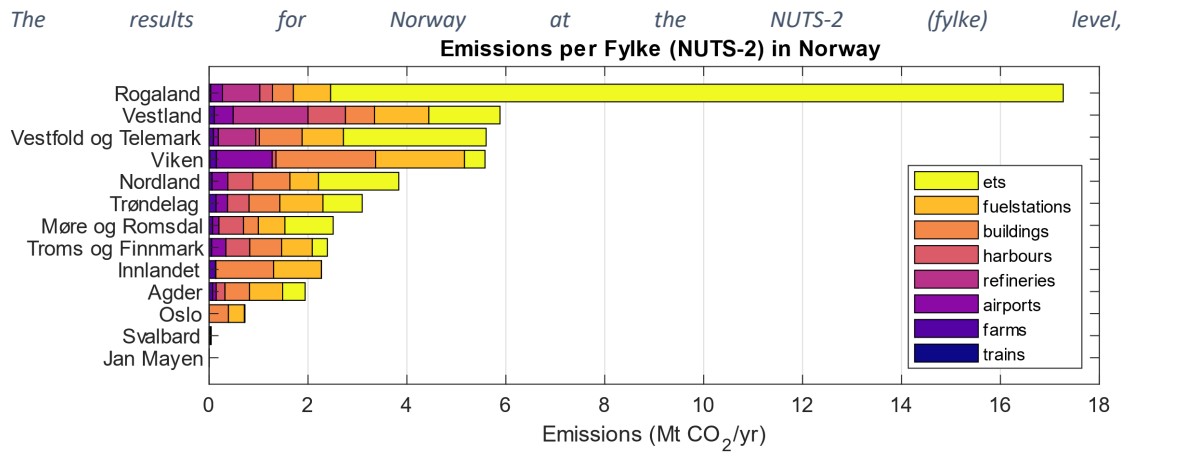


Figure 8) level show concentration and highlight the importance of industrial sources in Norway.
Rogaland fylke is the highest emitting. This is because in Stavanger, a city in Rogaland known as 'the
oil capital of Norway', in addition to reported emissions from petroleum facilities physically around
the city, many of the ETS-registered point source emissions from offshore facilities are legally
registered to company offices in Stavanger.

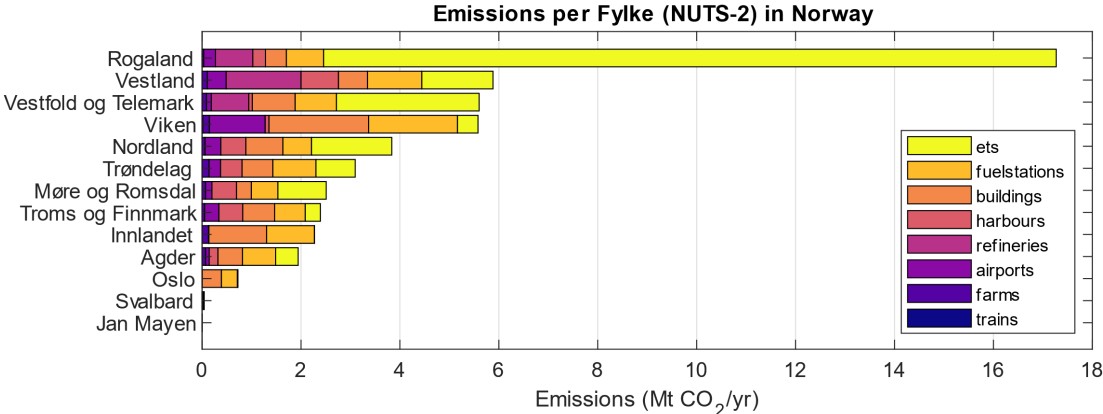


*Figure 8: CO2 emissions per NUTS-2 region (fylke) in Norway. The very high emissions in Stavanger (Rogaland) are driven largely by ETS-registered point sources. Stavanger is known as the oil capital of Norway. Note that Oslo fylke itself is small (ranked 11th), coextensive with only the heart of the city, and that Viken (ranked 4th) is the region which encompasses the greater Oslo region.*

Viken, the region of greater Oslo, has 5.8Mt of $CO_2$ emissions. The model results show that 32% of these emissions come from vehicles and 36% from buildings. Fossil fuel heating has been phased out of most buildings in Norway so these emissions are from light commercial activity, such as small burners, boilers, and generators not reporting to the ETS. A full 20% of emissions in Viken (1.1Mt) are associated with Norway's largest airport, the Oslo airport at Gardermoen. As described in the Methods, total emissions from aviation bunker fuel use in the country are allocated across airports in the country pro-rated by 2018 passenger volume. This approach could be biased and emissions from cargo flights, long-haul flights, and military aviation, should be located at airports different from those handling the most passenger traffic. This is a limitation of the current model.

Table 2 presents results at the municipality (*kommune*, or LAU-1) level for the top 20 municipalities. The relatively low emissions from the cities of Oslo (ranked 11th), Bergen (ranked 10th) and Trondheim (ranked 19th) is surprising given these are the three largest cities in Norway. Industrial emissions from ETS sources are the primary emissions drivers for the top four cities. The city-level results do also reveal some challenges with the model. The "refineries" category is defined as the residual between the national total emissions associated with industrial facilities and the total reported by the ETS facilities, and this residual is allocated evenly across facilities tagged as "refineries" in OSM. Overall this residual is small, but since there are few refineries, for individual cities it is substantial. Also noteworthy are the major emissions from harbors in the residential island archipelago of Øygarden. Currently emissions from marine bunker fuel are allocated evenly across all facilities tagged as "harbor" in OSM. In Øygarden there are many small-boat facilities, often not even selling fuel, yet at the same time the island region outside of Bergen is also heavily trafficked by large offshore work ships and cargo ships. Improving the methods use for spatializing emissions from marine bunker fuel use would help Improve the model for Norway and other countries with extensive marine traffic.

| Municipality (kommune) | Total | Airports | Buildings | ETS | Farms | Vehicles | Harbours | Refineries | Trains | TiOx |
|---|---|---|---|---|---|---|---|---|---|---|
| Stavanger | 12,109,439 | - | 149,270 | 11,779,396 | 4,935 | 146,650 | 28,932 | - | 256 | - |
| Porsgrunn | 2,079,447 | - | 17,446 | 1,989,186 | 441 | 67,040 | 4,822 | - | 512 | - |
| Sola | 1,395,161 | 208,654 | 23,320 | 1,100,663 | 448 | 37,710 | 24,110 | - | 256 | - |
| Tønsberg | 1,262,066 | - | 81,972 | 347,759 | 3,731 | 67,040 | 4,822 | 756,230 | 512 | - |
| Ullensaker | 1,223,520 | 1,128,279 | 29,898 | - | 1,981 | 62,850 | - | - | 512 | - |
| Haugesund | 1,202,557 | - | 17,292 | 1,133,338 | 1,015 | 46,090 | 4,822 | - | - | - |
| Øygarden | 1,088,329 | - | 37,224 | 67,910 | 2,695 | 79,610 | 144,660 | 756,230 | - | - |
| Sandnes | 905,490 | - | 56,100 | - | 980 | 92,180 | - | 756,230 | - | - |
| Alver | 864,906 | - | 31,174 | - | 9,198 | 58,660 | 9,644 | 756,230 | - | - |
| Bergen | 729,745 | 331,913 | 157,344 | 30,033 | 3,353 | 205,310 | - | - | 1,792 | - |
| Oslo | 724,800 | - | 386,628 | 10,468 | 2,002 | 322,630 | - | - | 3,072 | - |
| Sunndal | 694,376 | - | 8,008 | 670,648 | 3,150 | 12,570 | - | - | - | - |
| Karmøy | 616,538 | 27,177 | 20,218 | 442,562 | 413 | 58,660 | 67,508 | - | - | - |
| Bamble | 596,183 | - | 3,388 | 541,806 | 77 | 46,090 | 4,822 | - | - | - |
| Rana | 584,501 | 20,400 | 7,920 | 503,573 | 6,006 | 46,090 | - | - | 512 | - |
| Vefsn | 530,372 | 14,620 | 34,936 | 446,234 | 294 | 33,520 | - | - | 768 | - |
| Fredrikstad | 518,362 | - | 186,010 | 71,105 | 8,722 | 117,320 | 9,644 | - | 256 | 125,305 |
| Årdal | 467,475 | - | 2,288 | 456,373 | 434 | 8,380 | - | - | - | - |
| Trondheim | 458,851 | - | 233,640 | 45,422 | 2,289 | 167,600 | 9,644 | - | 256 | - |
| Senja | 451,891 | - | 27,962 | 304,611 | 266 | 41,900 | 77,152 | - | - | - |

*Table 2: Estimated CO2 emissions for 2018 for the top 20 emitting municipalities in Norway, as generated by OpenGHGMap.*

The model can be explored as tabular data, as a gridded raster model, or visualized on a map. Figure 9 provides an overview of the distribution of emissions across Norway, aggregated at the county and municipality levels. A concentration of emissions in Stavanger (in the southwest corner) and Porsgrunn (an industrial area in the south) is clearly visible.

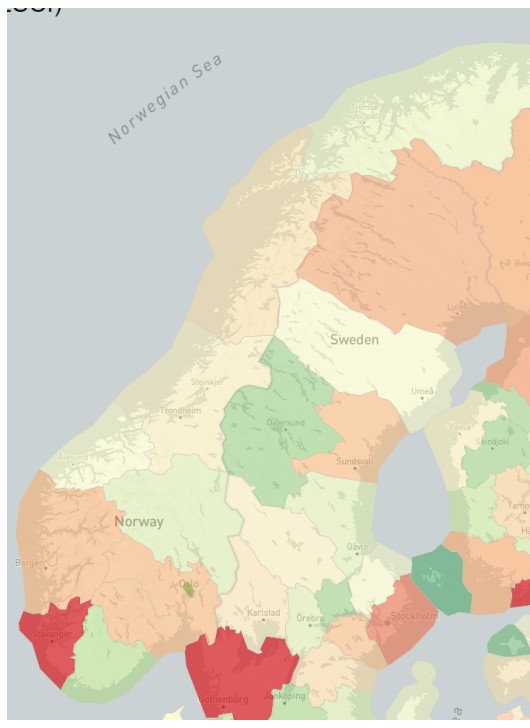
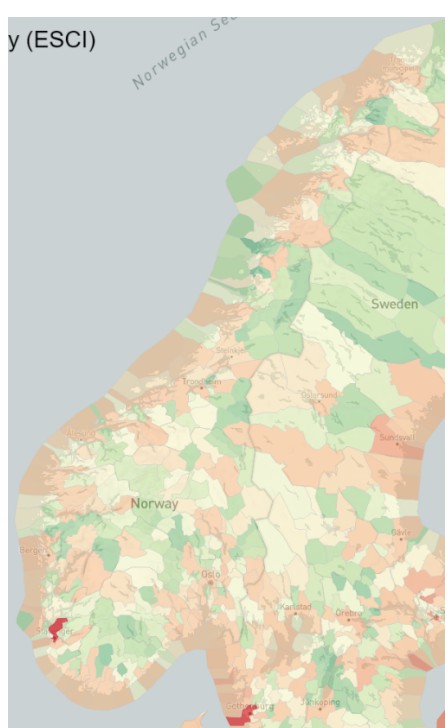

*Figure 9: Screenshot of the website heatmap visualization of OpenGHGMap-estimated CO2 emissions at the NUTS-2 county level (left) and municipality level (right) in Norway. Regions are color coded from green (lowest) to red (highest) emitting region in the country.*

Internally, the model attributes all national emissions to points across the country. It is possible to zoom in and view these emission point sources. Figure 10 provides a screenshot from the model visualization for the city of Trondheim, a city of 200,000 located in mid-Norway. The dots over each building, farm, fuel station, and ETS facility are scaled according to the estimated amount emissions

coming from that point. Orange dots show ETS-registered facilities. Purple dots in the figure show fuel
stations. The fine grey dots in the figure show all buildings registered in OSM. As detailed in the
Methods, emissions from several categories are allocated to buildings. The use of fossil fuel for
building heating is extremely rare in Norway. The emissions in the "building" category in Norway are
mostly from light commercial activity: boilers, generators, ovens, and the similar emissions from light
commercial activity which are below the ETS reporting threshold. As discussed above, it is difficult to
characterize buildings (e.g. buildings as different as a hospital, mall, auto body shop, and small cottage
are not distinguishable, nor can mansions be differentiated from cottages) (Milojevic-Dupont et al.,
2020), but this is clearly a frontier where further work is merited.

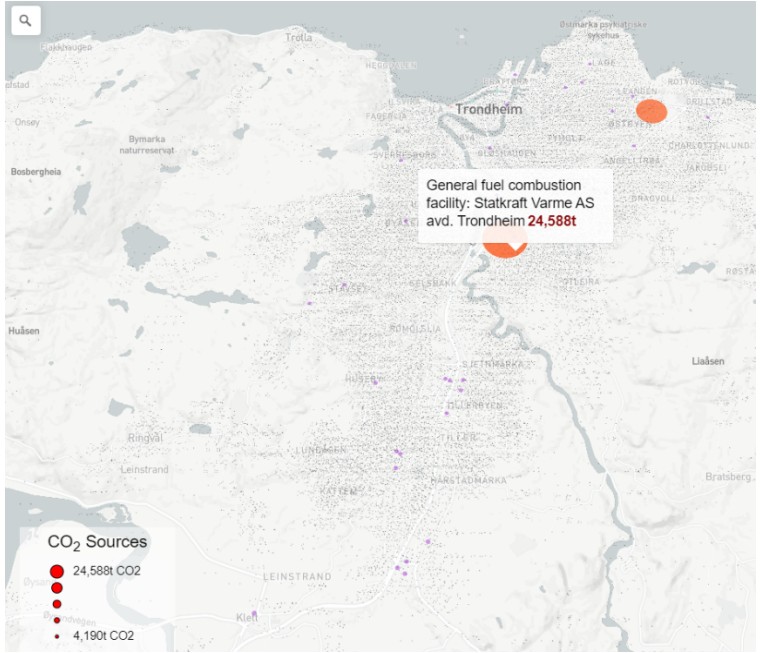

*Figure 10: Example visualization of spatialized CO2 emissions inventory for Trondheim, a city of 200,000 in mid-Norway, and*
*the surrounding region. Small grey dots represent individual buildings; purple dots are emissions from fuel stations, and the*
*large orange dots are ETS-registered point source facilities (a waste incineration plant and a factory making mineral wool).*
*This detailed view, while only an estimate, can provide residents and government agencies a thought-provoking view of what*
*decarbonization will look like for their town.*


## 5. Code Availability

The source code not available at the time of writing. The authors plan to clean up the code and prepare
a publicly usable version in the future. This will be linked at the Zenodo data repository and project
home page.

## 6. Data Availability

Datasets are available via Zenodo at https://doi.org/10.5281/zenodo.5482480 (Moran, 2021)
The Zenodo DOI is: 10.5281/zenodo.5482480
The model homepage, with an interactive map, is: https://openghgmap.net

## 7. Limitations, Uncertainties, and Future Work

One limitation of the approach presented in this paper, and a potential source of difficult-to-detect bias, could be inconsistent coverage in OpenStreetMap. As OSM is a crowd-sourced dataset there is no assurance of homogeneous coverage. Some areas of the country may be well-covered in OSM and others only sparsely (Hecht et al., 2013). This could introduce biases such as underreporting the number of fueling stations and thus underestimating vehicle traffic. The authors are not aware of any effort to characterize the consistency of OSM coverage; this would be a valuable next step both for the work presented here as well as for the OSM project and work derived therefrom.

For countries which do not participate in the ETS and do not have a similar domestic MRV system for large point source carbon emitters, spatializing emissions from point source polluters will be a challenge. Resources such as OSM and the Power Plant Database, which have considerable information at the facility level (e.g. output in megawatts and fuel source for power plants), could be of use.

The spatialization of emissions from vehicles and buildings - the two largest emissions categories - is challenging. The assumption in OpenGHGMap that every fuel station serves an equal volume and mix of vehicles is simplistic. The lack of even basic data characterizing buildings by height, area, age, or material, makes it impossible to differentiate buildings as varied as a terrace house block, separated house, mall, or hospital. Some novel approaches for characterizing building stocks have recently been proposed (Haberl et al., 2021; Milojevic-Dupont et al., 2020; Peled and Fishman, 2021) which could be used. Developing more accurate town-level models of building emissions may require different modelling approaches, such as utilizing data from national building cadaster registries or from advanced remote sensing datasets such as from synthetic aperture radar satellite constellations, airborne LIDAR sensors, and machine learning used with mobile airborne or ground cameras.

OpenGHGmap treats the CRF National Inventory Reports (NIRs) as authoritative. However, these inventories contain uncertainties. The NIR reports provide annexes which discuss uncertainties at the sector, sub-sector, and activity levels. The current version of the OpenGHGMap model does not exploit this uncertainty information, but future versions may. At the present time the OpenGHGMap focuses on spatially distributing the reported national emissions totals, and limits uncertainties to that spatialization exercise rather than including also the uncertainties within the NIR itself. Related to this it is noteworthy to mentioned related work on intercomparison of national emissions totals (Elguindi et al., 2020) and an assessment of uncertainty in the bottom-up EDGAR v6.0 model (Solazzo et al., 2021). Since OpenGHGMap treats national inventories as a fixed constraint with no uncertainty, the sources of uncertainty in the model are purely related to the spatialization of emissions. These uncertainties, and modeling choices, are discussed in the relevant section of Methods above.

Our emissions inventory can support local authorities in their journeys towards climate neutrality in multiple manners. The inventory can help make local and regional sources of emissions more tangible for diverse politicians, city administrations and local communities and provides a good starting point, especially for communities that lack a detailed GHG emissions inventory. Making an abstract concept such as greenhouse gas emissions more visible will enable discussions regarding localization and upgrading of facilities and infrastructures and will provide a basis for emblematic changes with high impact potential for the region. Connecting the inventory to digital urban twins with detailed information regarding built environment characteristics, may help overcome the current limitations of lack of building data.

In order to further develop the model, we will actively discuss and test it with local authorities to fine-tune it to their needs in order to make informed decisions. Furthermore, we will explore how we can further refine data collection, analysis and spatialization through the use of GIS combined with crowdsourcing and citizen science.

We foresee a number of use cases for the results presented here. For one, many local governments in Europe do not have an emissions inventory. The estimated inventory presented here presents a baseline initial estimate. This can be used to reveal which are the priority areas for reduction in each locale. For example, while vehicle electrification is highly promoted, it could be the case that for some regions emissions from residential or commercial buildings, or industrial sources are multiple times higher than from private cars and thus represent more important reduction opportunities. The results presented here are not a full replacement for an inventory prepared using a tool like the GHG Protocol for Cities. A bespoke inventory will be more detailed but the approach presented here can act as a starting point, help with classifying emissions and provide a benchmark against which estimates can be compared or even calibrated. The process of preparing the inventory itself usually triggers discussions about solutions. As the body of solutions grows it is possible to imagine cities soon able to construct a Climate Action Plan based on a menu of options. An estimated inventory like the one presented here could be used to prioritize or filter a longer list of solutions into the shorter set most suitable for each city. Finally, the results presented here have some communication value. There is much discussion about decarbonization at the national and EU level, but many are curious about what this should look like at their town, building, or business level. The results presented here can help people translate macro-level concerns into a more tangible vision of what should change in their home town, and how they can participate in that transition.

To conclude, we present a new European emissions inventory which disaggregates national $CO_2$ inventories to city and county level administrative jurisdictions. The model is broadly consistent with the ODIAC and EDGAR results but shows higher cell-level variability and provides results per-jurisdiction rather than in a gridded form. The estimated inventories provided by this model can help local governments begin establishing an emissions inventory.

## 8. Author Contributions

DM constructed the core model and led the manuscript writing. PP, HZ, HW, and JT contributed to the results analysis. KRG contributed to the introduction literature review, and conceptual framework. TW, AW contributed to the manuscript. HM, JK, DK, and AS contributed the aviation and marine emissions modules of the model.

## 9. Acknowledgements

This work was conducted with support from the Norwegian Research Council under grant # 287690/F20.

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
