# Peer review of "Estimating CO2 Emissions for 108,000 European Cities"

_Earth System Science Data, 2021_

## Author Response (AR1)

The high-level referee comments were:

- Both raised the idea of adding more discussion on how these results will match policy discussions. In response we have added a full paragraph to the Discussion. At the same time ESSD is a venue for papers which describe a model and dataset. This paper is primarily a data descriptor paper, not a discussion paper.
- Referee 2 pointed out that the national inventory reports are themselves not perfectly certain. This is an interesting consideration. In the current version of OpenGHGMap we limit concern to uncertainty regarding the spatialization of national emissions, assuming that the reported national totals are correct. In future updates it would be interesting to include consideration of that fact that the national inventories are themselves uncertain. We have added a discussion about this to the Uncertainties section of the manuscript.

The other comments were quite specific. We offer our point-by-point replies below:

**Reply to Referee 1**

My opinion of the technical work performed in the paper is good: an impressive work in data blending. Clearly, one could question and discuss every and single assumption: for instance, I was quite surprised by this one: "Legally, the control of the emissions is likely at the registered address, so there is sense in calling attention to emissions which are controlled from there." and even this one "Emissions from vehicles are apportioned equally to fuel stations as located in OSM" made me a bit unease, thinking to the time spent in the past in attributing meaningful traffic flows to road networks.

**REPLY: Fully agreed. It is possible to go much further into depth on almost any area. An interesting takeaway is that this type of emissions modeling is change-oriented research: the goal is to provide information to motivate and guide change. So it is in cases more appropriate to think about where change will / should occur, rather than where emissions physically occur. Our goal is to make a model which is tied to the physical reality of where emissions occur, but which can be also directly linked to intervention opportunities.**

Nevertheless, the authors have worked in a very transparent way, and all the proxies used are duly justified and accessible. Chapter 7 on the limitations and risks from their approach is overwhelmingly honest and clear. I fully share authors' worry about OSM coverage and reliability, as for many other crowd sourced datasets, and I would like to suggest the authors not stopping to test the use of other more "institutional" datasets (e.g., cadaster, road traffic data, etc.) to improve the robustness of their work. On summary, I think the results obtained are correct, in the sense they logically follow from the set of assumptions taken, a set that is one among many possible ones, chosen as a compromise among feasibility, meaningfulness and robustness.

On the contrary, I am not fully convinced by the use of the results the authors envisage: "Our emissions inventory can support local authorities in their journeys towards climate neutrality in multiple manners. The inventory can help make local and regional sources of emissions more tangible for diverse politicians, city administrations and local communities and provides a good starting point, especially for communities that lack a detailed GHG emissions inventory (lines 698-700)."

The experience from the GCoM and other local initiatives shows that local authorities need: 1) a precise estimation of the emissions they can influence by their policies and 2) tools helping them to evaluate the consequences of their actions. The dataset developed does not seem to offer this feature: for instance, attributing emissions equally to fuel stations does not "see" the effect of local traffic control measures. Even very important measures (such as e.g., closing the whole city to the internal combustion cars) would be extremely diluted across all fuel pumps of the nation, making difficult for the mayor to show benefits. Similarly for industries: the map shows several hotspots correspondence of the legal address of industries on which local authorities have presumably little influence.

In the results produced, both emissions manageable and unmanageable by local authorities are inextricably mixed and it is very difficult in some cases understanding how the information could support appropriate policies. In extreme cases, the approach taken could even push inappropriate measures: by absurd for instance, a mayor of a small town could be tempted of zeroing traffic related emissions in its jurisdiction simply closing a fuel station (or moving it to the neighbor town).

I would like to suggest the authors to reflect more on the possible uses of their results, and to present them in a more useful way. For instance, instead of showing results per macro-sectors, a finer or different subdivision could be more appropriate, or other solutions could be possible.

**REPLY: These are good comments. We share the referee's vision of a large body of future research to make emissions reporting more granular and actionable. While we fully agree with the referee's comment, it is not clear what specific and feasible changes to make to the current manuscript based on this response. Obviously adding more spatial and temporal detail and linking to policy tools would be a good further step. But realizing that would require a major new modelling effort and paper. The model and paper already represent a substantial advance for the field. We hope to have further improved results to present in the future.**

**Regarding the request to "reflect on possible uses" – this is primarily a data and model description paper, not a discussion paper. That said, we appreciate the referees request. We have added a paragraph to the Discussion:**

**"We foresee a number of use cases for the results presented here. For one, many local governments in Europe do not have an emissions inventory. The estimated inventory presented here presents a baseline initial estimate. This can be used to reveal which are the priority areas for reduction in each locale. For example, while vehicle electrification is highly promoted, it could be the case that for some regions emissions from light industry, buildings, or industrial sources are multipliers higher and thus represent more important reduction areas. The results presented here**

are not a full replacement for an inventory prepared using a tool like the GHG Protocol for Cities. A bespoke inventory will be more detailed and the process of preparing the inventory itself usually triggers discussions about solutions. As the body of solutions grows it is possible to imagine cities soon able to construct a Climate Action Plan based on a menu of options. An estimated inventory like the one presented here could be used to prioritize or filter a longer list of solutions into the shorter set most suitable for each city. Finally, the results presented here have some communication value. There is much discussion about decarbonization at the national and EU level, but many are curious about what this should look like at their town, building, or business level. The results presented here can help people translate macro-level concerns into a more tangible vision of what should change in their home town, and how they can participate in that transition.»

On summary: the methodology chosen by authors to distribute national emissions down to the finer jurisdiction is surely one of the many possible ones and overall I think it is correct and transparently justified. But I am not sure it is the most appropriate for guiding local authorities in their decision on emissions control.

**REPLY: We agree that methodological innovation in this field is needed. We present just one possible approach to distribute national emissions to the jurisdiction.**

**As to which method is most appropriate for guiding local authorities in their decision on emissions control, this will be revealed as different local authorities make their choices about conducting an inventory, using this tool, or using other tools.**

Specific comment: please consider changing the terminology used e.g., in the title: there are not 108,000 "cities" in Europe. See e.g https://ec.europa.eu/eurostat/web/nuts/tercet-territorial-typologies

**REPLY:**
**We've revised the abstract to further clarify. It now reads:**
    **Here we present a new CO2 emissions inventory for all 116,572 municipal and local government units in Europe, containing 108,000 cities at the smallest scale used. The inventory spatially disaggregates the national reported emissions, using 9 spatialization methods to distribute the 167 line items detailed in the UN's Common Reporting Framework. The novel contribution of this model is that results are provided per administrative jurisdiction at multiple administrative levels, following the region boundaries defined OpenStreetMap, using a new spatialization approach.**

**Regarding our choice for the title, "cities" is catchier than "local administrative and jurisdictional divisions" and quickly conveys the sense of what we present in the results. In the abstract and Introduction we are quite clear from the outset about what administrative divisions are used so it is not the case that the paper is misleading.**

Minor: Line 102: IPCC not IPCCC
**REPLY: Thank you; fixed.**

**Reply to Referee 2**
This study represents valuable modelling work as a first attempt on estimating the $CO_2$ emissions from cities across Europe. The authors do a great job in identifying and mapping the emission (point) sources and their $CO_2$ emissions, and the methodology is clear and well explained, a very difficult task given the current state of the proxies' availability and the incomparability between countries administrative units. As an exercise, the authors compare their estimates with data from two global inventories (ODIAC and EDGAR v6.0) and include the Norwegian case-study.

In the current context of the Paris Agreement and reduction pledges, countries should not base their reductions only on national level inventories but take into account cities' individual contributions. In this context, this study is a valuable piece of information to be shared with local authorities. However, the authors should be careful on the sort of concrete messages they want to pass on to the cities. Would be of interest to add a section on how authors see their findings as contributing to local reductions (something like result-mitigation format) and which would, in their vision, be the most critical activities to challenge, in terms of CO2 reduction.

**REPLY: We agree this is an interesting topic for exploration, but in this ESSD publication we prefer to keep a tight focus on a clear description of the data and underlying model. The discussion proposed merits a fuller treatment, including probably a closer comparison between the city-level inventory tools like GHG Protocol for Cities, which is very action-Oriented, and top-down models like this an EDGAR.**

**To further respond to this comment we have added a new paragraph to the Discussion:**

**"We foresee a number of use cases for the results presented here. For one, many local governments in Europe do not have an emissions inventory. The estimated inventory presented here presents a baseline initial estimate. This can be used to reveal which are the priority areas for reduction in each locale. For example, while vehicle electrification is highly promoted, it could be the case that for some regions emissions from residential or commercial buildings, or industrial sources are multiple times higher than from private cars and thus represent more important reduction opportunities. The results presented here are not a full replacement for an inventory prepared using a tool like the GHG Protocol for Cities. A bespoke inventory will be more detailed but the approach presented here can act as a starting point, help with classifying emissions and provide a benchmark against which estimates can be compared or even calibrated. The process of preparing the inventory itself usually triggers discussions about solutions. As the body of solutions grows it is possible to imagine cities soon able to construct a Climate Action Plan based on a menu of options. An estimated inventory like the one presented here could be used to prioritize or filter a longer list of solutions into the shorter set most suitable for each city. Finally, the results presented here have some communication value. There is much discussion about decarbonization at the national and EU level, but many are curious about what this should look like at their town,**

**building, or business level. The results presented here can help people translate macro-level concerns into a more tangible vision of what should change in their home town, and how they can participate in that transition.»**

As mentioned below in the specific comments, I miss a paragraph on uncertainty calculation. The UNFCCC NGHGIs include in their NIRs (Annexes) uncertainties reported for each sector, sub-sector and activities. Solazzo et al., 2021 estimated as well uncertainties for EDGAR v5.0, year 2015. Even if uncertainty calculation is still not implemented in ESCI, I would strongly advise the authors to include some reference values for uncertainties from the above mentioned sources, as an idea of magnitude.

**REPLY: Good point. We have renamed section 7 to "Limitation, Uncertainties, and Future Work" and added the following paragraph:**

**OpenGHGmap treats the CRF National Inventory Reports (NIRs) as authoritative. However, these inventories are contain uncertainties. The NIR reports contain annexes which discuss uncertainties at the sector, sub-sector, and activity levels. The current version of the OpenGHGMap model does not exploit this uncertainty information, but future versions may. Related to this it is noteworthy to mentioned related work on intercomparison of national emissions totals (Elguindi et al., 2020) and an assessment of uncertainty in the bottom-up EDGAR v6.0 model (Solazzo et al., 2021). Since OpenGHGMap treats national inventories as a fixed constraint with no uncertainty, the sources of uncertainty in the model are purely related to the spatialization of emissions. These uncertainties, and modeling choices, are discussed in the relevant section of Methods above.**

Overall the paper is well written and has a clear structure. The authors highlighted as well the limitations and future work, in a transparent and honest way, and to which I also contribute with some ideas (see specific comments).

I recommend it for publication, subject to addressing the changes and suggestions, as highlighted above and below in the specific comments.

Specific comments:

Title: A bit confusing (108,000 cities) given the number in the abstract.

**REPLY: We've clarified this so the abstract now reads: "Here we present a new CO2 emissions inventory for all 116,572 municipal and local government units in Europe, containing 108,000 cities at the smallest scale used. The inventory spatially disaggregates the national reported emissions, using 9 spatialization methods to distribute the 167 line items detailed in the UN's Common Reporting Framework. The novel contribution of this model is that results are provided per administrative jurisdiction at multiple administrative levels, following the region boundaries defined OpenStreetMap, using a new spatialization approach."**

Line 23: the authors refer to ESCI as "inventory", I would introduce here the ESCI model name and rename it as "modelled estimates" or "modelled city inventory".

**REPLY: Good suggestion. We've changed that to read: «Here we provide a new pan-European model estimating emissions at the municipality level»**

Line 24: nine instead of 9

**REPLY: Fixed**

Line 24-25: "UN's Common Reporting Framework" Reading all the manuscript, I would be consistent and call it everywhere UNFCCC Common Reporting Format (CRF).

**REPLY: Fixed, at all instances in the paper**

Line 27: please add Zenodo link after "is available at...."

**REPLY: Fixed. Added Zenodo DOI.**

Line 33: European climate goals" I would add here references to the new target documents, Green Deal etc.

**REPLY: We respectfully prefer not to name specific programs since these are changing very quickly. For example, the now-flagship Mission on 100 Climate Neutral Cities was a much lower profile project when we drafted the paper, and it is likely that the flagship policy programs will continue to evolve quickly in the future. The general phrase "climate goals" seems adequate.**

Line 40: delete "the" C40 and add comma after C40, C40 and GCoM are two different entities.

**REPLY: Fixed**

Line 49: no one = none

**REPLY: We meant it as written, to emphasize the fact that no particular one of these is ideal.**

Line 53: Would be great to add a short paragraph about ongoing EU projects (CoCO2, PAUL)

**REPLY: Thank you for pointing us to these projects, we have added mention of them in the paragraph.**

Line 70: Next to what you list, top-down approaches also use inventories like EDGAR as proxies

**REPLY: Sorry, we did not understand what change to make based on this comment. We have clearly highlighted the EDGAR model already.**

Line 75: EDGSR is not a global top-down emission data product, it is a tier 1 bottom-up inventory providing as well gridded products. I think this sentence needs rephrasing

**REPLY: Thank you; fixed. We've edited the manuscript so that EDGAR is described as a "global emissions data product" not a "top-down global emissions data product".**

Also, please consider adding everywhere in the text EDGAR v6.0

**REPLY: We now indicate EDGAR v6.0 when it is first introduced, and at several more points in the paper.**

Line 89: Worth mentioning hot-spot detection satellite studies for CO2 city/megacities emissions (OCO-2, OSSEs) also INFLUX experiments (USA)

**REPLY: We had already cited INFLUX and OCO-2, and we have added a citation for OSSE.**

Line 102: IPCC delete one C, and I think there is a confusion between IPCC and UNFCCC CRFs, please be aware that the CRFs you are referring to are not IPCC but UNFCCC. If you want to reference IPCC, then you have to mention the methodology for reporting according IPCC, 2006 and IPCC Refinement 2019. But again, reading the paper, you use the sectors according UNFCCC CRFs.

**REPLY: Thanks; this is fixed throughout to refer to UNFCCC.**

Line 103: The number is again inconsistent with the title and the abstract. Please consider referring to one unit (administrations, municipalities, cities etc.)

**REPLY: Thanks for spotting. Fixed.**

Line 106: after "existing model" please consider adding some examples of models

**REPLY: Thanks, good suggestion. We replaced "existing models" with "EDGAR and ODIAC".**

Line 109: please add EDGAR v6.0 and reference the datasets

**REPLY: Fixed. We now refer to EDGAR v6.0 throughout.**

Line 111: There three datasets are not models, consider naming it datasets, inventories

**REPLY: Fixed. Changed to "datasets".**

Line 113: I would rephrase as: "Additionally, our model is targeted to be useful to citizens and local governments, at city level, by identifying the sources of their city's CO2 emissions"

**REPLY: Thank you; good suggestion. We have used this better phrasing.**

Line 115: emission attribution

**REPLY: Good suggestion. Taken.**

Line 156: Again, it is not the IPCC CRFs you are looking at, is the UNFCCC CRFs

**REPLY: Fixed (same as above)**

Line 167: in the context of LULUCF emissions I would include references to the a) to e) points (e.g. Grassi et al, 2018, Petrescu et al., 2020 AFOLU, etc.)

**REPLY: Good suggestions; we've added these.**

Line 264: I would suggest using GPS information (e.g. TomTom) as done in the Carbon Monitor (https://www.nature.com/articles/s41597-020-00708-7)

**REPLY: We were aware of this; we've followed this suggestion and now mentioned it the manuscript as a good idea for future work.**

**The TomTom data for nowcasting emissions is probably very powerful. For the time being we are satisfied with the approach of localizing vehicle emissions at fuel stations rather than on roads, but it would be good in future work to be able to do even this more accurately (to distinguish between fuel station types and sales volumes).**

Line 326: How is this allocation comparable with the IPCC methodology? international flights are not reported to national inventories. You perhaps overestimate here?

**REPLY: We edited the manuscript to read:**

**«Total emissions associated with kerosene used for aviation fuel (the sum of the CRF categories "Fuel combustion in domestic aviation (1.A.3.A)" and "International aviation (1.D.1.A)") reported by EU member states and calculated compliant with IPCC 2006 guidelines (Maurice et al., 2006). These emissions are attributed to airports proportionally to total passenger kilometers (pkm).»**

**In the case of the European Union, international aviation emissions were reported but not included in the national emission accounts. The reported items can be inspected, e.g., here: https://www.eea.europa.eu/data-and-maps/data/national-emissions-reported-to-the-unfccc-and-to-the-eu-greenhouse-gas-monitoring-mechanism-17 The aviation component of OpenGHGMap is based on NTNU's AviTEAM project, part of NTRANS (https://www.ntnu.no/ntrans). That project focuses heavily on spatial estimation of aviation emissions.**

Line 326: How is this allocation comparable with the IPCC methodology? international flights are not reported to national inventories. You perhaps overestimate here?

**REPLY: We edited the manuscript to read:**

**«Total emissions associated with kerosene used for aviation fuel (the sum of the CRF categories "Fuel combustion in domestic aviation (1.A.3.A)" and "International aviation (1.D.1.A)") reported by EU member states and calculated compliant with IPCC 2006 guidelines (Maurice et al., 2006). These emissions are attributed to airports proportionally to total passenger kilometers (pkm).»**

**In the case of the European Union, international aviation emissions were reported but not included in the national emission accounts. The reported items can be inspected, e.g., here: https://www.eea.europa.eu/data-and-maps/data/national-emissions-reported-to-the-unfccc-and-to-the-eu-greenhouse-gas-monitoring-mechanism-17 The aviation component of OpenGHGMap is based on NTNU's AviTEAM project, part of NTRANS (https://www.ntnu.no/ntrans). That project focuses heavily on spatial estimation of aviation emissions.**

**In the current model we allocate aviation bunker fuel emissions to airports. Military flights are not included in this, so our estimate of flight-related emissions may be an underestimate.**

Line 350: If you use locations of farmland from CORINE, why don't you use it for retrieving information on AD (forest area) or use FAO FRA AD? You could only select the pixels belonging to city administration areas.

**REPLY: Yes, good idea. The bigger challenge though is that "Agriculture and forestry" are lumped together in the emissions inventory. Adding the forest locations as you suggest would just distribute the same total across more locations (farms plus forests). Probably, this would distribute the forestry emissions across large forest areas even when only a small area is being logged. Also, forestry is presumably the minority share of the "agriculture + forestry" combination, so it makes more sense to prioritize spatializing the agricultural emissions.**

**We will definitely take this suggestion on board for future refinement.**

Line 358: not CRF report, but UNFCCC CRFs

**REPLY: Fixed (same as above).**

Line 368: same as for aviation, international shipping should not be included if compared to UNFCCC reported numbers

**REPLY: The shipping emissions model in OpenGHGMap is taken directly from http://www.smartmaritime.no/**

**As from aviation, all marine bunker fuel emissions from the CRF inventory were allocated to activity across the country.**

**An additional paragraph has been included to clarify that international shipping is indeed part of national GHG inventories:**

**«Emissions from the maritime sector are part of international bunker fuel emissions together with international aviation. In both cases, emissions are calculated as part of the national GHG inventories but not included in national totals.»**

Line 419: the title here is a bit confusing, ((Land Use, Forestry, and Stock Change) and you also mention waste. First you refer to sector 5: CRF Table 5 (not section), then you describe the category 4 (should be CRF Table 4) and then you get back to 5 again. I would open the discussion with a general sentence on the sinks from both Tables 4 and 5, and then detail on each sector.

**REPLY: We changed the heading to "Land Use, Forestry, Stock Change, and Waste (Short-cycle carbon)". We've rewritten the opening sentence of the section to better clarify; it now reads: «Our model is focused on reporting $CO_2$ emissions from fossil fuel combustion and industrial processes. We explicitly set aside so-called "short cycle carbon", that is, carbon which is already in the biosphere stock. We limit the model to focus on emissions of carbon taken from the fossil stock.»**

Line 425: you can use everywhere the acronym LULUCF

**REPLY: we mostly use the acronym but sometimes use the full text.**

Line 444, 449: EDGAR v6.0 inventory

**REPLY: Addressed above.**

Line 453-460: I understand you could not use the methodology of EDGAR in the first phase of writing the paper, but for review I would strongly suggest to contact M. Crippa and rewrite this paragraph adding the explanations regarding EDGAR v6.0 AD and methodology.

**REPLY: Respectfully, this is a big ask from the referee. Adding this component is asking several weeks of work from us, yet would not change the substantive contribution of this paper. This is a dataset/model description paper which clearly presents our model. We have already presented a validation exercise and discussed key points of uncertainty and comparability. Asking us to go into a detailed point-by-point method comparison with a quite different model is a big undertaking. Papers which go into depth comparing multiple models are rare, for a reason – it is a big job, and usually one that is tackled by dedicated papers, not the papers which initially present a model.**

**If there were an easy way to respond to this comment, we would do so. We have already presented a high-level results and method comparison with EDGAR. Respectfully, we would argue that asking for even more depth would be asking too much from this paper.**

Line 471: space after 6.0

**REPL: Thanks. Fixed typo.**

Line 473: Only here you introduce the EDGAR v6.0. Please consider doing this in the beginning of the study

**REPLY: Fixed (same as above).**

Line 475: The three models/inventories

**REPLY: Fixed. Changed to "modeled inventories"**

Line 477: would authors consider adding a table with categories covered by each dataset? Would help identifying shortcomings related to emission differences

**REPLY: We agree that this is conceptually a good idea, but again here the referee suggest we do work in this model-specific paper which is more suited to a dedicated model-intercomparison paper.**

Line 486: Please complete the units t CO2 / year , per cell?

**REPLY: Fixed (added "t CO2/yr")**

Line 505: Figure 2 caption: I would rephrase: "ESCI reports higher cell-level variability ranging from $10^1$ to $10^5$ t $CO_2$/yr, while ODIAC reports most cells in the range of $10^2$-$10^4$ t $CO_2$/yr." Also, please delete JRC (consistency purposes) and replace model to inventory or dataset.

**REPLY: thank you; we have used this suggestion. We removed JRC and replaced all instances of "model" with "dataset" in the caption.**

Line 511: delete "Then we", "(ie. by city)" and "We compared results both" Rephrased should be: "Next, we converted the administrative region definitions from ESCI to a raster map compatible with the EDGAR v6.0 and ODIAC gridded datasets and we compared the results aggregated by administrative level across the models, at the city level (i.e. ....)."

**REPLY: Thank you; we have used this suggestion.**

Line 514: Figure 3, X axis: a) and b) per city-level instead of per region? c) and d) per country instead of per region?

**REPLY: Yes, this would be a possible change, but we believe the axis labels are clear already.**

Line 518: Here, as mentioned in the general review above, I would add a section on Uncertainties and review the values reported by UNFCCC NIRs and Solazzo et al 2021. Just to give a range of uncertainty to some of the activities.

**REPLY: Addressed above.**

Line 537 Figure 4: It is a nice figure but the legend is confusing, I would suggest instead of adding "a" in green and black, to draw a green and black rectangular (same as the cities names) around the CO2 and GHG and color them as well. As ESCI only simulates CO2 emissions, should be green, please change the caption with: "Comparison between ESCI results (green) and the community level emissions inventories of 44 European cities (black)...."

**REPLY: We've clarified the caption so it now reads: "Figure 1: Comparison between OpenGHGMap results and the community level emissions inventories of 44 European cities. Color coding is used to indicate whether the city self-reports CO2 or GHG (CO2eq) emissions. Since OpenGHGMap reports only CO2 emissions this is limited to an indicative comparison, not a precise comparison."**

Line 542: Please add a reference value, per capita or per 100,000 inhabitants for CO2 emissions, for someone who is not aware of city CO2 emissions levels, is not clear if 11kt is a high or low value.

**REPLY: We've edited the text to state "11ktCO2 for a population of 150000 is unrealistically low»**

Line 561: Please add colored gradient  legends and units to be able to read these figures !. Add CO2 emissions to the caption.

**REPLY: Fixed. The cation now reads: "Figure 2: OpenGHGMap.net website screenshots. CO2 emissions per municipality in absolute terms (left panel) and per capita terms (right panel).  Darker colors (browns, purples) indicate higher emissions (absolute values can be found at the website http://openghgmap.net)."**

Line 572 as a general comment: to be able to better compare between countries, perhaps in the next version of the model you could classify/rank the municipalities by inhabitants, 0-1000 villages, followed by towns (small, medium, large), cities and megacities (over a million).

**REPLY: Yes, good idea. This is also possible to do using the Zenodo dataset SI accompanying this paper.**

Line 596 Figure 8: I see three times figure 8, is this the same or there is some difference between them? Add CO2 emissions to caption.

**REPLY: Apologies, this was a formatting problem in the document. Caption is fixed as suggested.**

Line 599: please delete "level" after Figure 8)

**REPLY: Apologies; this was a formatting issue in the PDF. It is fixed.**

Line 634: caption Table 2: Estimated CO2 emissions

**REPLY: Indeed. Fixed.**

Line 640: Figure 9: please consider refining the figure by clearing the text appearing on the upper left corner and add gradient colored legend with units for the CO2 emission. Add CO2 in the caption after "ESCI-estimates"

**REPLY: Caption fixed and changed to read "Figure 9: Screenshot of the website heatmap visualization…" to make it clear this is a screenshot of a portion of the website, not a standalone figure. We do plan to add a gradient color legend to the website, but the current color scheme is very intuitive so in our opinion these figures are legible as-is.**

Line 656: Figure 10, please move the explanatory box on the left side of the orange dot to be able to see its size. Add CO2 after "spatialized"

**REPLY: Fixed caption. The label box is translucent so we think the orange dot underneath is clear enough that this figure does not need to be redone.**

Line 681: About the OSM coverage, you would perhaps consider, as I already mentioned, the GPS TomTom service for traffic

**REPLY: Yes, good idea, we will definitely consider adding this.**

Line 685: you model could be enriched by including satellite information (OCO-2, GOSAT, OSSEs, Plume Monitoring Inversion Framework (PMIF) studies) for the detection of hot-sport point sources (refineries, power plants at the outskirts of the cities).

**REPLY: Yes, definitely. The introduction contains a discussion of how statistics models such as this one could be combined with empirical data feeds from satellites. We look forward to collaborating on such projects.**

Lines 698-700: Would be nice to have a small section with a more detailed view of the authors on how their results could feed into local support practices, with some concrete

examples, critical sectors and mitigation action. Also, I think local city councils should be able to provide cadaster schemes for an overview of building types and activities.

**REPLY:**

**We have added a new paragraph to the discussion in response to this comment.**

**In this ESSD submission we have focused on presenting the model and dataset. We leave a fuller discussion on next steps to a future paper. We agree that the referee raises interesting questions, but we have already presented a valuable, self-contained advance in this paper and feel this paper stands alone with its current scope. Based on the referee comments it seems a full new discussion would be a major extension of the paper rather than filling a shortcoming of the current manuscript.**